# Automatically disambiguating medical acronyms with ontology-aware deep learning

Marta Skreta [1,2,3,4✉], Aryan Arbabi[1,2,3,4], Jixuan Wang [1,2,3,4], Erik Drysdale[5], Jacob Kelly [1,4], Devin Singh[1,3] & Michael Brudno[1,2,3,4✉]

Modern machine learning (ML) technologies have great promise for automating diverse clinical and research workflows; however, training them requires extensive hand-labelled datasets. Disambiguating abbreviations is important for automated clinical note processing; however, broad deployment of ML for this task is restricted by the scarcity and imbalance of labeled training data. In this work we present a method that improves a model's ability to generalize through novel data augmentation techniques that utilizes information from bio-medical ontologies in the form of related medical concepts, as well as global context information within the medical note. We train our model on a public dataset (MIMIC III) and test its performance on automatically generated and hand-labelled datasets from different sources (MIMIC III, CASI, i2b2). Together, these techniques boost the accuracy of abbreviation disambiguation by up to 17% on hand-labeled data, without sacrificing performance on a held-out test set from MIMIC III.

[1] Department of Computer Science, University of Toronto, Toronto, Canada. [2] DATA Team & Techna Institute, University Health Network, Toronto, Canada. [3] Centre for Computational Medicine, The Hospital for Sick Children, Toronto, Canada. [4] Vector Institute for Artificial Intelligence, Toronto, Canada. [5] The Hospital for Sick Children, Toronto, Canada. ✉email: martaskreta@cs.toronto.edu; brudno@cs.toronto.edu

Health care practitioners typically abbreviate complex medical terminology when preparing clinical records, saving time of writing out long terms/phrases, while making the text clear to an experienced professional in the context. Correctly disambiguating medical abbreviations is important to build comprehensive patient profiles, link clinical notes to ontological concepts, and allow for easier interpretation of the unstructured text by practitioners from other disciplines. Expanding abbreviated terms into their long forms is nontrivial since abbreviations can have many expansions. For example, "ra" can mean right atrium, rheumatoid arthritis, or room air depending on both its local (adjoining words) and global (type of note and other information in it) context. While disambiguating abbreviations is typically simple for an expert in the field, it is a challenging task for automated processing, which has been addressed by a number of methods going back at least 20 years. These methods largely rely on supervised algorithms such as Naive Bayes classifiers trained on co-occurrence counts of senses with automatically tagged medical concepts in biomedical abstracts[1]. Semi-supervised approaches also took root during this time: Pakhomov et al. improved the contextual representation of senses in clinical notes by augmenting them with text from the Web and biomedical abstracts, but were only able to validate their methods on eight abbreviations[2].

Modern methods that disambiguate abbreviations rely on the local context of the abbreviation to discern its meaning. A number of supervised machine learning (ML) models have been built for abbreviation disambiguation in medical notes, including ones based on support vector machines (SVM), Naive Bayes classifiers, and neural networks[3–6]. More recently, abbreviation disambiguation models have been fine-tuned using contextualized embeddings generated from BERT and ELMo model derivatives[7,8]. However, the development and deployment of methods for automated abbreviation disambiguation are limited by the availability of appropriate training data. Creating hand-labeled medical abbreviation datasets to train and test ML models is costly and difficult, and to the best of our knowledge, the only such publicly available dataset with training data and labels is the Clinical Abbreviation Sense Inventory (CASI)[9], which contains just 75 abbreviations. The sparsity of these datasets makes methods built based on them vulnerable to overfitting and inapplicable to abbreviations not present in the training data. This is evident in studies where training and testing models on different corpora can result in performance drops of 20–40%[5,6]. Moreover, the same studies typically disambiguate 50–2000 abbreviations, compared to >80,000 medical abbreviations that are in AllAcronyms, a crowd-sourced database of abbreviations and their possible expansions[10].

Finley et al.[11] utilized reverse substitution (RS) to auto-generate training data by replacing expansions with their corresponding abbreviations. For example, the phrase "Patient was administered intravenous fluid" in the training data was transformed to "Patient was administered ivf", and the label for this instance of the abbreviation "ivf" was "intravenous fluid". RS, however, creates imbalanced training sets because the distributions of terms in their abbreviated and long forms are often different. Some phrases, due to their obvious meaning, or because they are too long, are rarely written out fully; for example, milligrams (mg) next to medication dosage, or "in vitro fertilization" (see Results, below). Although additional work has improved RS by hand-labeling of specific instances[6], none of the existing methods for abbreviation disambiguation scale to tens of thousands of medical abbreviations listed in resources such as AllAcronyms.

An additional problem with medical abbreviation disambiguation is that the local context of a word is not always sufficient to disambiguate its meaning. For example, "rt" could represent "radiation therapy" or "respiratory therapy", and the phrase "the patient underwent rt" cannot be disambiguated without further information. Huang et al. showed that words can be better represented by jointly considering their local and global contexts[12], and Kirchoff and Turner demonstrated that document contexts are useful in medical abbreviation disambiguation tasks[13]. A study by Li et al.[14] represented acronyms in scientific abstracts using the embeddings of words with the highest term frequency–inverse document frequency (TF-IDF) weights within a collection of documents. This was motivated by the idea that acronym expansions are related to the topic of the abstract and that topics can be described by words with the highest TF-IDF weights.

In this work, we tackle the problem of disambiguating medical abbreviations in the absence of training data for any specific abbreviation, thus dramatically increasing the ability of such models to generalize to new texts. We took the following three-pronged approach:

1. We used information from related medical concepts to create more balanced and representative examples of training data for RS approaches. We did this by sampling sentences of related concepts in the immediate vector space and adding them to our training cohort, which is especially beneficial for medical concepts that are rare or not written in the training text.

2. We leveraged structural relationships in biomedical ontologies such as the unified medical language system (UMLS) to pre-train our models[15] by constraining medical concepts to be in the same vector space as their neighbors.

3. We defined a simple global context that combines medical knowledge from the entire note and used it in conjunction with the local context of an abbreviation to further improve the accuracy of abbreviation disambiguation.

Using these three techniques, we achieve an overall 17% improvement on CASI. Using automatically generated testing samples from i2b2, a collection of patient discharge summaries[16], we show a 2% accuracy improvement on i2b2, and over a 7% increase on abbreviations with little training data. Finally, we recruited medical professionals and students to hand label abbreviations in i2b2. We tested our model on these abbreviations and found a 16% improvement compared to the baseline.

## Results

**Disambiguation algorithm.** An overview of our approach to disambiguating medical abbreviations is shown in Fig. 1a. Our framework leverages the RS paradigm, but introduces a data augmentation technique that supplements the training set with samples of closely related medical concepts to reduce the false prior of training sets generated using RS and eliminate the need for labeling abbreviation datasets by hand. First, we learn word embeddings for terms in clinical notes by training a FastText model on MIMIC III notes[17]. We then map medical concepts in UMLS to the resulting vector space to generate a word embedding for every medical concept. Then, for a given abbreviation, we augment the training samples for each expansion with sentences containing closely related medical concepts determined using embedding distance (Fig. 1b). The key assumption is that sentences containing long forms of related medical concepts can be used as proxies for unknown abbreviations. For example, sentences containing the phrase "assisted fertilization" are more similar to sentences that contain the term "ivf" which expands to in vitro fertilization rather than to "ivf" expanding to intravenous feeding. Hence, they can be used as training samples for the concept "in vitro fertilization" even though we never see the

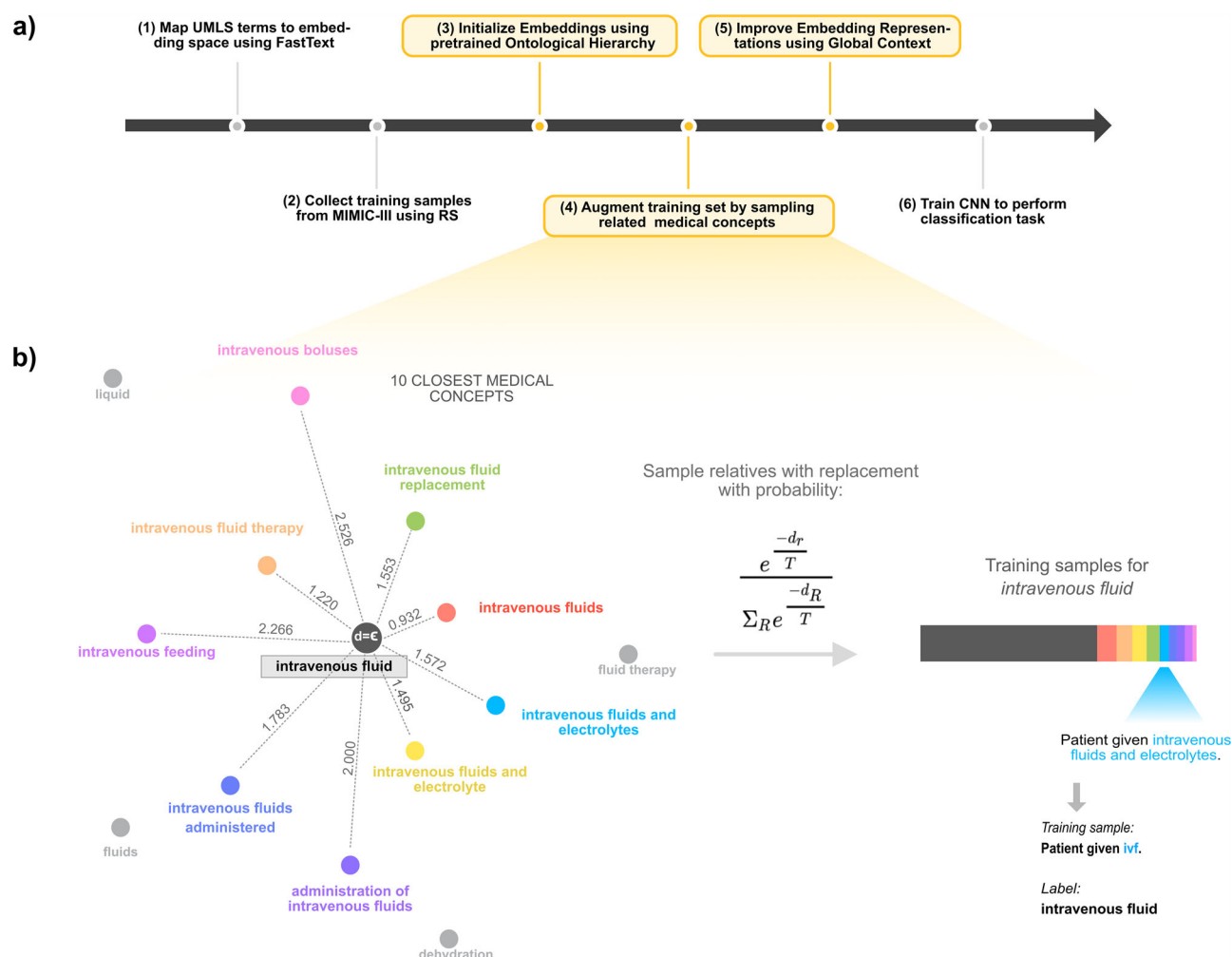

**Fig. 1 Overview of our abbreviation disambiguation pipeline for data collection and model training. a** Overview of our method. Our key contributions are indicated with yellow boxes. **b** Illustration of data augmentation technique for the training set. For each expansion, we sample sentences for the ten closest medical concepts using reverse substitution (RS) with a probability proportional to their Euclidean distance in the embedding space. The Euclidean distance is shown above the dotted line connecting the expansion to its relative. The probability of sampling is indicated above the arrow. $d_r$ is the Euclidean distance between the expansion and relative and $R$ refers to the ten closest medical concepts. During training, we learn a temperature $T$ using Bayesian optimization that is used in the sampling function. In the event that an expansion is present in the training corpus, we sample it with a distance of $\epsilon$, which we set to 0.001. We add each sample to our training set by replacing the relative with the abbreviation and using the target expansion as the label. An example of this is shown below the color bar. The color of each relative corresponds to the color in the bar to the right of the arrow, which reflects the proportion of the training set composed by that relative.

phrase "in vitro fertilization" in the training corpus. This method does not require expert annotation and thus scales to previously unseen abbreviations.

Using this training set, we trained a convolutional neural network (CNN) to perform the classification task of predicting the correct expansion for an abbreviation given its local context (the neighboring words) and global context (the whole note, represented by IDF-weighted word embeddings). Further details on each step are provided in the "Methods" section.

**Evaluation**. To evaluate the contribution of each component of our model, we compared its performance to models trained without the critical sub-components. The first model (Control) uses training samples acquired using RS without any alterations. The second (sampling with replacement, SWR) is similar to the first, but samples training sentences with replacement such that each expansion has an equivalent number of training samples. The third model (Relatives) incorporates our novel data sampling technique by including relatives of expansions in the biomedical

ontology (UMLS) into the training set. We sample concepts with replacement so that all expansions have an equivalent number of training samples. To evaluate whether using the structure of the ontology can improve the results we initiated it with weights learned from the hierarchical training task (+ hierarchical pretraining, HP). We also trained the models using both only the local neighborhood of the abbreviation (default) and by incorporating global context information for each sample (+ global). We used bootstrapping to obtain the mean for each abbreviation by resampling our predicted values and true values 999 times.

We consider two forms of accuracy. Micro accuracy is the total number of abbreviations correctly disambiguated divided by the total number of samples in the test set across all abbreviations with two or more possible expansions. Macro accuracy is the average of individual abbreviation accuracies and gives a better reflection on the performance of imbalanced datasets. A Wilcoxon signed-rank test was used to compare the macro accuracy results of different models (micro accuracy is a point estimate).

**Table 1 Micro and macro accuracy of (%) of our model on (a) CASI abbreviations, (b) i2b2 generated by hand labeling, (c) i2b2 generated by RS, and (d) MIMIC III generated by RS.**

| Dataset generation method | (a) CASI accuracy | | (b) i2b2 hand-labeled accuracy | | (c) i2b2 RS accuracy | | (d) MIMIC-III accuracy | |
|---|---|---|---|---|---|---|---|---|
| | Hand-labeled | | Hand-labeled | | RS | | RS | |
| | Macro | Micro | Macro | Micro | Macro | Micro | Macro | Micro |
| *Sampling method* | | | | | | | | |
| Control | 0.672 | 0.673 | 0.702 | 0.682 | 0.869 | 0.850 | 0.948 | 0.917 |
| Control + global | 0.686* | 0.687 | 0.738 | 0.745 | 0.877* | **0.862** | 0.955* | 0.929 |
| SWR | 0.705* | 0.708 | 0.701 | 0.680 | 0.864 | 0.834 | 0.948 | 0.914 |
| SWR + global | 0.715* | 0.712 | 0.701 | 0.677 | 0.873* | 0.850 | 0.956* | 0.931 |
| Relatives | 0.813* | 0.806 | 0.833* | 0.795 | 0.873 | 0.827 | 0.945 | 0.910 |
| Relatives + global | 0.825** | 0.820 | 0.855** | 0.816 | 0.886** | 0.842 | 0.954** | 0.925 |
| **Relatives + global + HP** | **0.841***** | **0.834** | **0.859** | **0.825** | **0.889***** | 0.848 | **0.961***** | **0.935** |
| Clinical BERT | 0.648 | 0.643 | 0.602 | 0.591 | 0.824 | 0.788 | 0.917 | 0.871 |
| Clinical BERT + Relatives | 0.721**** | 0.717 | 0.690**** | 0.699 | – | – | – | – |

*$p < 0.05$ (one-sided Wilcoxon signed-rank test compared with Control model).
**$p < 0.02$ (one-sided Wilcoxon signed-rank test compared with Relatives model).
***$p < 0.01$ (one-sided Wilcoxon signed-rank test compared with Relatives + global model).
****$p < 0.03$ (one-sided Wilcoxon signed-rank test compared with Clinical BERT model).
We sample training data with replacement (SWR) and augmentation with related medical concepts (Relatives). We report results for when we incorporate the ontology during pretraining (HP) and the global context of the note (global). Bolded values indicate the best-performing model for each column. We have omitted running ClinicalBERT + Relatives on the RS datasets due to computational constraints.

We evaluated our model on four datasets (see "Methods" for further detail):

(1) a held-out test set consisting of RS samples of abbreviation expansions from MIMIC III (20% of the dataset),

(2) an orthogonal dataset of 65 abbreviations from CASI with gold-standard annotations,

(3) 1116 abbreviations from i2b2 generated by finding sentences with expansions from AllAcronyms using RS, and

(4) 24 abbreviations from i2b2 hand-labeled by medical students.

Table 1 shows the micro and macro accuracies of our concept embedding model using our data augmentation technique on test sets from MIMIC-III, CASI, and i2b2. The first two columns show results on test sets generated by hand, while the last two columns show results on RS-generated datasets. While both types of datasets show improvements using our techniques, we note that the gain is stronger for hand-generated datasets and the performance on RS-generated datasets is higher overall. This is likely due to biases inherent in RS-derived test sets that more closely resemble the training set.

**CASI—hand-labeled dataset**. The p-values and performance differences between all models are displayed in Fig. 2. Training abbreviation with both local and global contexts gave significantly better performance than training on local context alone. Likewise, augmenting the training set with related medical concepts resulted in a 14% ($p = 1.2e{-}03$) increase in accuracy on CASI compared to the control. Incorporating global context increased this value to 15% ($p = 5.3e{-}05$) and hierarchical pretraining improved it by another 2% ($p = 1.4e{-}05$). This demonstrates that the global context in which related terms appeared and hierarchical information aided disambiguation.

While the main goal of this paper is to evaluate the data augmentation and the use of ontological information for pretraining, which can be applied to any method, as an additional baseline we downloaded and installed the codebase of Finley et al.[11] (https://github.com/gpfinley/towards_comprehensive) and utilized it on the CASI dataset. We found that our baseline model

("Control") has a 3% improvement (67% versus 64% for Finley et al.).

Figure 3 is a histogram displaying the performance difference between our best model (Relatives + global + HP) and the control model for CASI abbreviations. Notably, the performance improved for 38 out of 65 abbreviations. The abbreviations that benefited most were the ones where an expansion did not appear in the training corpus or appeared at a very low frequency. For example, our model increased the performance for the abbreviation "na" by 75% compared to the control. This is because "narcotics anonymous" (a possible expansion) only appears twice in MIMIC III. Incorporating related concepts such as "alcoholics anonymous" and "nicotine use" enabled us to create a better representation.

**i2b2—automatically generated dataset**. Table 1c shows the raw performance numbers and Fig. 2c displays the p-values and performance differences of the various model iterations tested on the larger test set of 1116 abbreviations using another orthogonal dataset, i2b2, with labels generated using RS. of all the models. While the full model outperformed the control, the total performance gain was more modest (2%). Higher overall performance and smaller improvement indicate that i2b2 more closely resembles MIMIC III with respect to the frequency of different disambiguations. For example, in the case of "ivf", there are significantly fewer instances of fully spelled out "in vitro fertilization" compared to "intravenous fluids" in both MIMIC III (zero versus 2503) and i2b2 (2 versus 49). At the same time, "in vitro fertilization" is the more common expansion in CASI (294 versus 181). This could indicate either a difference between the datasets, or human behavior: the RS method relies on the long form of an abbreviation to be written out fully, and this may be less likely with abbreviations that are either clearer in the context, or longer and hence rarely written out.

Further breaking down our model's performance on the i2b2 test set, we see that it performed better on abbreviations with little training data (Fig. 4). Almost a quarter of the 1116 abbreviations have less than 200 training samples per expansion on average. Our model performed 7% better than the baseline on these abbreviations.

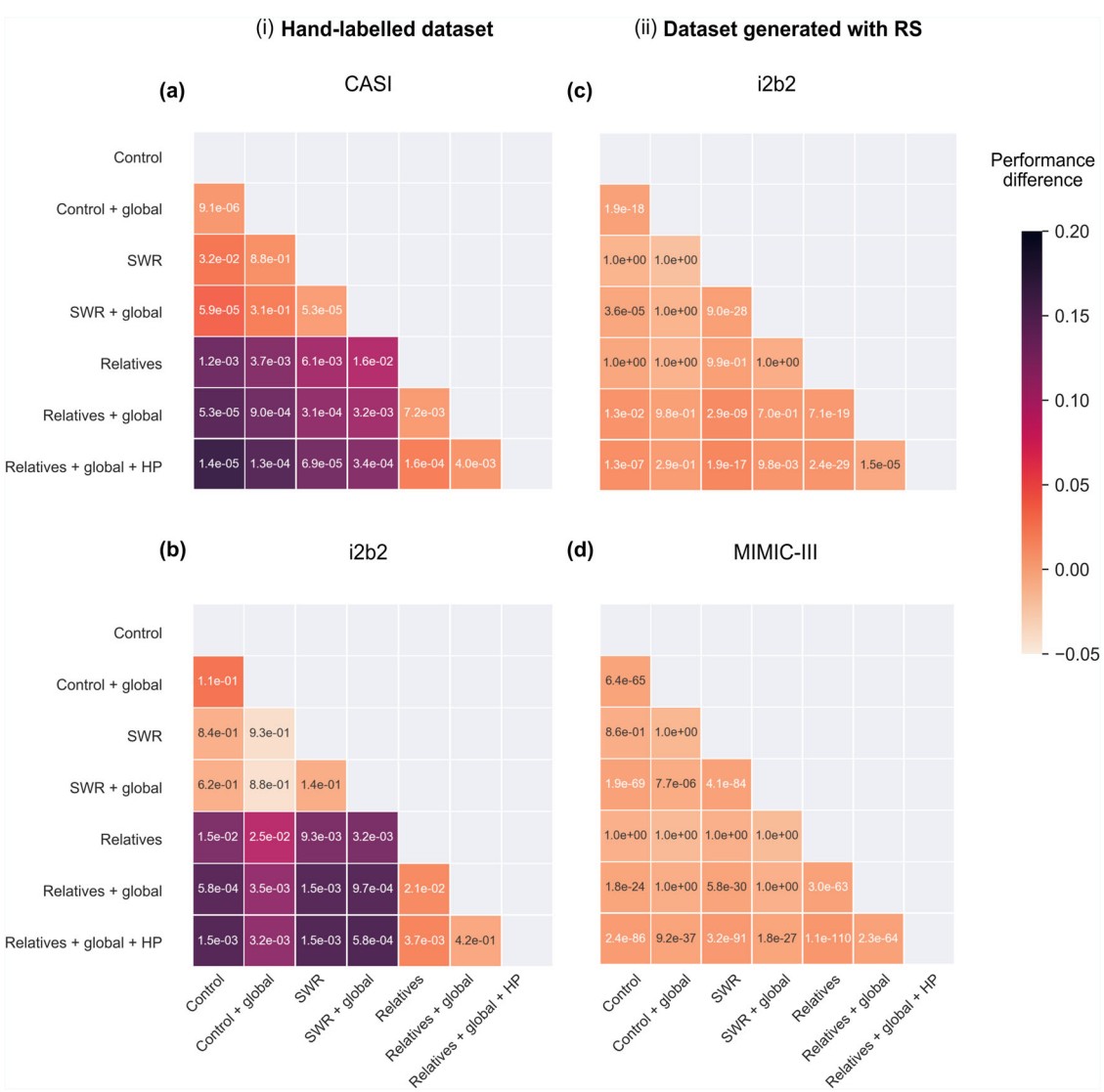

**Fig. 2 Matrix showing performance differences and p-values between all models on test sets generated by (i) hand-labeling and (ii) RS.** The external test sets are **a** CASI, **b** i2b2 generated by hand labeling, **c** i2b2 generated by RS, and **d** MIMIC-III generated by RS. The color intensity of each square reflects the performance difference between the corresponding model on the vertical axis and the model on the horizontal axis. p-Values were obtained using a one-sided Wilcoxon signed-rank test and are displayed inside each square. Note that we expect (and observe) no improvement from introducing relatives in the MIMIC-III dataset (**d**), as the default (control) algorithm better represents the underlying data distribution in MIMIC-III.

**i2b2—hand-labeled dataset.** As an orthogonal metric, we manually labeled a dataset of 270 abbreviations from the i2b2 corpus. Of the 270, only 24 had multiple expansions, illustrating the strong bias in which abbreviations are written out fully (see "Methods"). Figure 2b shows the performance difference between our model and the baseline as well as the p-values between all models. Table 1b shows the raw performance. We find that our model generalizes better to the test set by almost 16% compared with the control model.

**Contextualized embeddings.** Impressive advances have been made in a variety of natural language processing (NLP) tasks by learning context-dependent embedding representations[18,19]. One advantage of using such embeddings is that a single word can have a different embedding depending on its context. Specifically, attention-based models such ClinicalBERT have shown dramatic improvements in clinical NLP tasks[20]. We fine-tuned the ClinicalBERT model for our abbreviation disambiguation task and compared it to our baseline CNN model (see "Experiments" for training specifications). We find

that in our case, using embeddings from ClinicalBERT does not offer any improvement on our baseline. We believe that this is because the clinical notes in both the training and test sets lack structure typical of training corpa for BERT such as Wikipedia and that the average number of expansions per abbreviation is small enough that a single embedding per term is sufficient. The benefit of using a simpler model such as our baseline is that BERT models are memory intensive; because we train one model per abbreviation, this adds up quickly and becomes impractical.

To evaluate whether our data augmentation technique (Relatives) is applicable to more complex models, we applied clinicalBERT to the CASI dataset and find that there is a significant gain in performance (~7%, from 64.8% to 72.1% for macro accuracy, from 64.3% to 71.7% for micro accuracy). We have also observed a significant performance increase for the i2b2 hand-labeled dataset (8.8% gain for macro accuracy, 10.8% gain for micro accuracy). This demonstrates that our novel data augmentation technique can be useful regardless of the underlying model architecture.

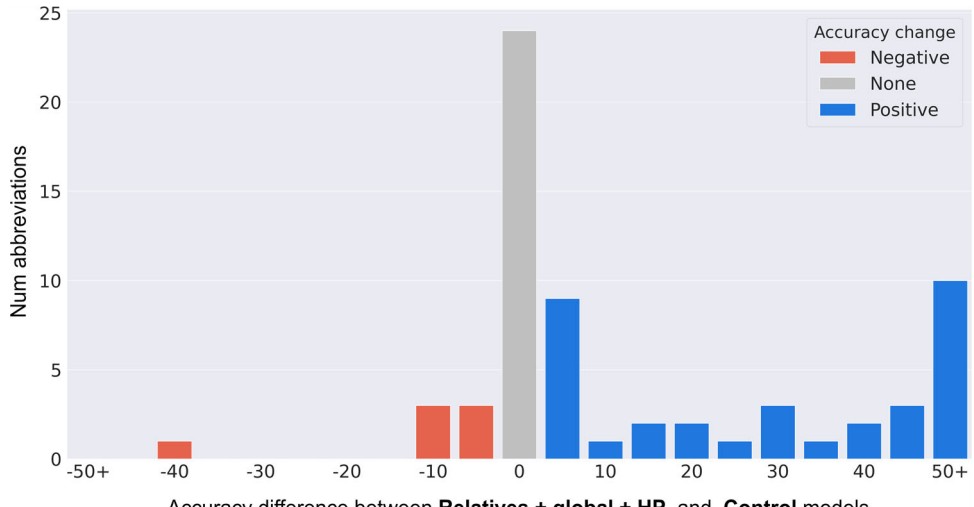

**Fig. 3 Histogram showing the accuracy difference between the Relatives + global + HP model and control model performance (%) on CASI abbreviations.** The x-axis shows bins of 5%, where each unit is the bucket means (i.e., $x = 0$ bins data from an accuracy difference of $-2.5\%$ to $+2.5\%$). The abbreviations that improved using our method ("Positive") are shown in blue, the abbreviations that had a performance decrease are shown in red ("Negative"), and the abbreviations that were unchanged are in gray ("None").

**Downstream task—predicting medical test**. We tested the impact of expanding abbreviations in clinical notes on the prediction of required medical tests in an Emergency Department setting. We developed a model that extracts UMLS concepts in clinical notes from the Hospital for Sick Children (Concept Unique Identifiers, CUIs) and predicts whether the patient received a specific clinical test (forearm X-ray, abdominal ultrasound, urine testing, and bloodwork). We found that without expanding abbreviations, the model achieved an accuracy of 78.09% on an independent test set. After training the model on clinical notes with expanded abbreviations, the performance increased to 78.51% ($p < 0.05$ using the standard bootstrap method), showing that there is value in disambiguating abbreviations in clinical notes before using them in downstream tasks, at least in this specific setting. See "Methods" for additional details on this experiment. This work will be more fully described in a separate paper.

## Discussion

One of the key requirements of most ML methods is the presence of datasets that are used to train the model. In the medical space, such training datasets are often difficult and expensive to obtain. Particularly within the problem of disambiguating abbreviations, the largest human-labeled datasets that are available contain dozens of abbreviations[3], while in the literature, we know there exist tens of thousands[21]. Utilizing methods that utilize RS typically leads to biased training datasets and overfitting by the algorithm.

In this work, we demonstrate a general algorithm for disambiguating medical abbreviations that scales to previously unseen medical acronyms by utilizing biomedical ontologies as prior medical knowledge. Our approach is based on the ideas introduced by us in an extended conference abstract[22], where we explored the global context and data augmentation to improve performance for this task. This manuscript, however, goes beyond the preliminary work by introducing the use of hierarchical ontologies as a pretraining step, improving the Bayesian Optimization algorithm to select an optimal temperature, creating and analyzing the hand-labeled dataset as another orthogonal metric, tripling the set of analyzed abbreviations, and enhancing the "Results" section by including the comparison to Clinical BERT,

and Finley et al., as well as including the analysis of the downstream clinical task (test prediction). Overall, our approach overcomes the lack of training data and achieves additional improvements by considering the relationship of the terms in the ontology, introducing a pre-training step to help embed concepts. For all samples, we are also able to generate better representations by considering the global context in which an abbreviation appears. Because of these improvements, our overall framework demonstrates up to 17% higher accuracy of abbreviation disambiguation on auxiliary datasets.

One notable limitation of our approach is runtime. As was done in the previous work[6], we train one model per abbreviation; the training cost of this is significantly more expensive using our data augmentation technique than the baseline model since we do 25 rounds of Bayesian optimization to search for the optimal temperature (although this can possibly be reduced, we did not experiment with that in this study). To give a concrete example, training 65 abbreviations (the size of CASI dataset) on a single Tesla V-100 GPU takes ~25 h using our data augmentation technique but ~1 h for the baseline. Furthermore, given the variability among the various datasets used in this study, more work is required in creating a unified corpus for possible abbreviations and related medical terms. While AllAcronyms give an idea for what senses are possible, there exist senses in CASI that are not in AllAcronyms. Finally, to make the pipeline fully end-to-end, better abbreviation detection models should be developed. While some publically available models do exist, they are trained on a small fraction of all possible medical abbreviations[5].

Our approach has immediately led to better results for the abbreviation disambiguation problem and has further implications for the development of other ML-based methods. Utilizing examples of closely related concepts from an ontology has already shown improved results for named entity recognition[23] and word sense disambiguation in biomedical texts[24]. We believe that such approaches can be useful for addressing a wide variety of biomedical problems.

## Methods

**Datasets**. We used seven datasets in this study, all of which are publicly available:

(1) We used clinical notes from MIMIC III as our training set. We collected sentences from MIMIC III containing abbreviation expansions, as well as

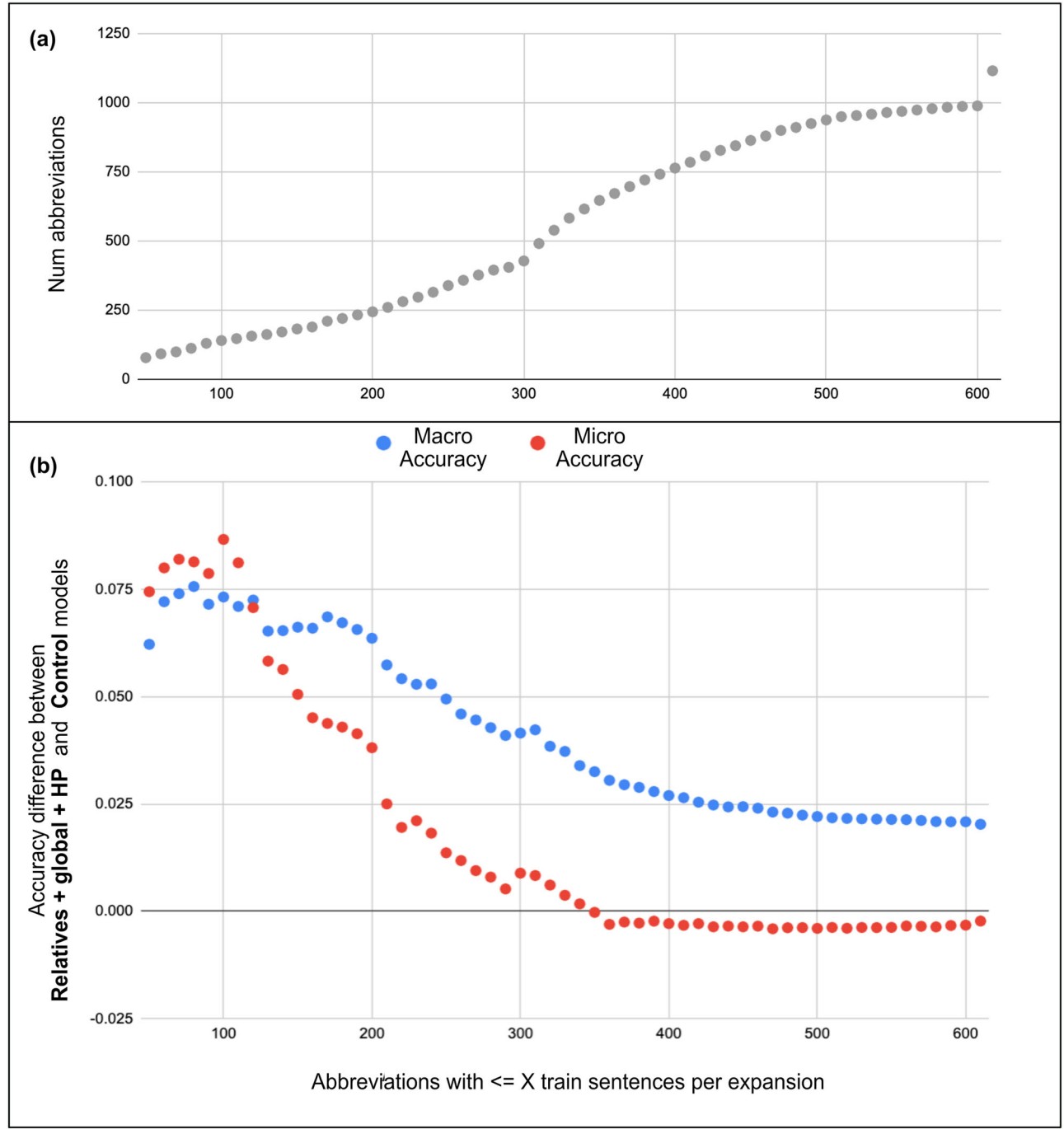

**Fig. 4 Performance difference between Relatives + global + HP and control models on i2b2 test with an increasing number of training samples per abbreviation.** The x-axis marks abbreviations with an average less than or equal to X training sentences per expansion. **a** Number of abbreviations that have an average of less than X training sentences per expansion. **b** Difference in macro (blue) and micro (red) accuracy for these abbreviations between our best and control models.

concepts in UMLS to augment our training set. We also used MIMIC-III to pre-train word embeddings using FastText and IDF weights.

(2)   We augmented our training sets based on relationships between expansions and concepts defined by UMLS Metathesaurus.

(3)   We used the medical section of AllAcronyms, a crowd-sourced database, to obtain a list of 80,000 medical abbreviations and 200,000 potential expansions. We removed abbreviations with only one disambiguation and those that do not appear in UMLS, resulting in 30,974 abbreviations.

(4)   We used the CASI dataset as an orthogonal test set to measure model generalizability. We removed expansions that are the same as the abbreviation (for example, the word "it" is an abbreviation and an expansion) and abbreviations with one expansion since the disambiguation

task is trivial in that case. This left us with 65 abbreviations. On average, each abbreviation has 4 expansions with 459 test sentences.

(5)   As another test set, we used i2b2. This dataset does not have hand-labeled annotations, so we used RS to generate labels. There are 1116 abbreviations in i2b2 containing more than one expansion. On average, each abbreviation has 4 expansions with 97 test sentences.

(6)   To ensure the in-distribution performance was not compromised with our augmentation techniques and demonstrate the level of overfitting, we tested our model on a small test set from MIMIC III. We test our model on the same 1116 abbreviations as in the i2b2 test set.

(7)   We also generated a hand-labeled dataset to better reflect the frequency of abbreviations used in practice. Starting with 270 abbreviations whose

expansions occurred with a similar frequency in i2b2, we sampled up to 50 sentences containing each abbreviation. We developed a website that presented this sentence and possible expansions (Supplementary Fig. 1) to one of seven volunteer medical students at the University of Toronto. Each abbreviation was hand-labeled by two students. If the labellers disagreed it was reviewed by a staff physician (D.S.). We started with 270 abbreviations (Supplementary Data 1); however, after labeling only 24 abbreviations had more than one expansion (Supplementary Table 1).

**Word embeddings**. To map semantically similar words close in vector space, we trained word embeddings in an unsupervised manner on the MIMIC-III corpus using FastText. FastText learns word embeddings by representing each word as a bag of character $n$-grams[17]. This is useful for creating good representations of rare words in the training corpus since it considers sub-word information and for predicting embeddings of concepts not present in the training corpus. We joined multi-word medical concepts from UMLS with a "_" symbol to represent and embed them as a single token.

**Training set sampling**. For each expansion for a given abbreviation, we augmented the training samples with the ten most related medical concepts. Figure 1b illustrates our sampling technique. The degree of relatedness was measured as the Euclidean distance between the expansion phrase and UMLS concepts. We sampled each relative (with replacement) in proportion to its distance from the expansion according to the following probability:

$$p_{\text{sampling}} = \frac{e^{-d_r/T}}{\sum_R e^{-d_R/T}} \qquad (1)$$

where $d_r$ is the Euclidean distance between the expansion and the relative and $T$ is the temperature of the distribution. $R$ refers to the 10 closest medical concepts. If sentences for an expansion were present in the training corpus, we treated the expansion as a relative with a distance of $\varepsilon$ (a hyperparameter which we set to 0.001). The temperature $T$ is a "sharpening" function[25]. For each abbreviation, we searched for a temperature that minimizes the loss on the MIMIC III validation set using Bayesian optimization, constraining the temperature to be between $2^{-1}$ and 2. We found that smaller values overfit to MIMIC III, while larger ones added too much noise. For each abbreviation, we performed 25 iterations of Bayesian optimization using the Tree-structured Parzen Estimator algorithm and took the model with the lowest validation loss[26].

**Sentence embeddings**. We mapped an input sentence to a vector representation using a simple encoder similar to that used by Arbabi et al.[23]. The network consists of one convolution layer with a filter size of one word, followed by ELU activation[27]. Max-overtime pooling was used to combine the output into a single vector, $v$

$$v = \max_t(\text{ELU}(W_1 x^{(t)} + b)) \qquad (2)$$

where $x^{(t)}$ is the word embedding of the term at index $t$. $W_1$ and $b$ correspond to the weight matrices and bias vectors, respectively, which we learned through training.

A fully connected layer with ReLU activation followed by L2 normalization was used to map $x$ to the final encoded sentence representation:

$$e = \frac{\text{ReLU}(W_2 v)}{||\text{ReLU}(W_2 v)||_2} \qquad (3)$$

The embedded sentence is a representation of the local context. To incorporate the global context of a sample, $g$, we took the weighted average of the embedding vectors for each word in the document. The embeddings were weighted using IDF weights trained on the MIMIC III corpus. The vector $g$ was calculated as follows:

$$g = \frac{\sum_{i=1}^d u_i * w(t_i)}{\sum_{i=1}^d w(t_i)}, i \neq j \qquad (4)$$

where $j$ is the index of the abbreviation, $i$ is the index of the $i$th word in the document, and $d$ is the number of words in the document. $u_i$ is the word embedding and $w(t_i)$ is the IDF-weighting of the $i$th word.

We then concatenated $g$ with the encoded sentence vector, $v$, and normalized it to produce the final encoded sample embedding

$$e = \frac{\text{ReLU}(W_2[v;g])}{||\text{ReLU}(W_2[v;g])||_2} \qquad (5)$$

*Classification using a CNN*. Our model was trained to minimize the distance between a target expansion embedding and its context (Fig. 5).

Our model represents expansion embeddings with an embedding matrix, $H$, where each row, $H_c$, corresponds to the embedding of an expansion for a given abbreviation. To do the classification task of assigning an expansion label, $c$, to an input sentence, $e$, we took the dot-product of $H$ and $e$ and apply a softmax

function, such that

$$p(c|e) = \frac{\exp(H_c e)}{\sum_{c'} \exp(H_{c'} e)} \qquad (6)$$

We labeled the abbreviation with the expansion having the largest probability $p(c|e)$.

**Pre-training using ontological relationships**. Ontologies are structured medical terminologies that link related concepts together. Incorporating structure in this form can entangle embeddings of related concepts and generate more refined embedding clusters in the medical domain[23]) and has been shown to improve vector space representations for general language tasks[28].

We linked related concepts together using a hierarchical medical ontology as a prior. The benefit of this is that concepts with insufficient training data are constrained to be close to their relatives. For an abbreviation, we took all expansions and closest medical concepts within a Euclidean distance of $\delta$ (we treated $\delta$ as a hyperparameter and found $\delta = 2.6$ to work best). We linked these concepts using the lowest common ancestor in the UMLS hierarchy and trained a model to predict which concepts from UMLS best fit the context. This is similar to our abbreviation model; however, to take structural information into account, we first learned a matrix, $H'$, where each row, $H'_c$, is the raw concept embedding for a given concept, $c$. $H_c$ is then derived by taking the sum of $H'_c$ and all of $c$'s ancestors' embeddings, $H'_{P_c}$, such that

$$H_c = H'_c + \sum_{P_c} H'_{P_c} \qquad (7)$$

The ancestors of $c$ project $c$ to a global location, while the raw embedding of $c$ learns a local location. During training, we backpropagated through $H'_c$ and $H'_{P_c}$ by the same amount. Thus, if a concept's ancestors are not present during training, their embeddings still get updated. If the concept itself is not present during training, it simply inherits the embeddings of its ancestors. We then used these weights to initialize our model for the abbreviation disambiguation task.

**Contextualized embeddings**. We compare our embeddings to contextualized word embeddings generated from ClinicalBERT[20]. In general, we find that ClinicalBERT performs significantly better than the original BERT model for this application. We finetune the ClinicalBERT model on our task by stacking two hidden layers joined by a nonlinear activation function which takes as input the embedding from the ClinicalBERT model and outputs the probability of each expansion. We experiment with the number of hidden layers and find that two work best. BERT-style architectures output an embedding for each token in the input, as well as a "start of sentence" token that serves to encode the entire sentence[18]. We experiment with using the "start of the sentence" token and the abbreviation token, and find that the "start of sentence" token performs better. We also tune other hyperparameters such as sentence length, batch size, and learning rate.

**Downstream task—medical test prediction**. We acquired 176,140 clinical notes from the emergency department at the Hospital for Sick Children as well as the corresponding codes for tests that the patients received. The possible tests were: [C-arm, CT, Cecostomy Maintenance, ECG, GI Assessment, MRI, Other, PICC Maintenance, Tube Maintenance, Ultrasound, X-Ray, None]. First, we detected abbreviations in the clinical notes using the CARD abbreviation detection model developed in ref. [5]. While this model is publically available, it was trained to detect only 500 medical abbreviations, which is a small proportion of all existing abbreviations and may have missed abbreviations in our notes. 156,801 sentences (89%) were found to contain at least one abbreviation using the CARD model. We removed all sentences that did not contain any abbreviations (11%), since we are comparing the performance of expanded vs. non-expanded sentences. We then expanded the detected abbreviations using our pipeline. We then extracted CUIs from both the original and expanded notes and encoded them using a one-hot representation. We trained a simple neural network consisting of two fully connected layers linked by a ReLU nonlinear activation function that took in the one-hot-encoded CUIs and predicted which of the tests the patient should have received. The model was evaluated based on whether the corresponding tests were actually ordered by the physicians.

## Experiments

*Model training*. We trained our model on sentences from MIMIC III. We collected sentences containing expansions from CASI and medical concepts from UMLS using RS. In total, 105,161 concepts in UMLS were found in MIMIC III. To learn word vectors, we trained a FastText model using a skip-gram architecture with an embedding size of 100[17]. For the classification task, we built one model for each abbreviation. To train our model, we used a maximum of 1000 samples per expansion and found a context window of 3 words to work best. On average, each abbreviation had 3.46 expansions. We trained our models on 60% of the sample set, validated it on 20%, and kept 20% as a held-out test set. We trained all concept embedding models for 100 epochs with a learning rate of 0.01 and saved the epoch with the lowest validation loss. We ran the Bayesian Optimization acquisition function using 15 random seeds and used the one with median validation loss to get typical model performance. Only 1 random seed was used for the i2b2-RS dataset due to extremely long runtimes.

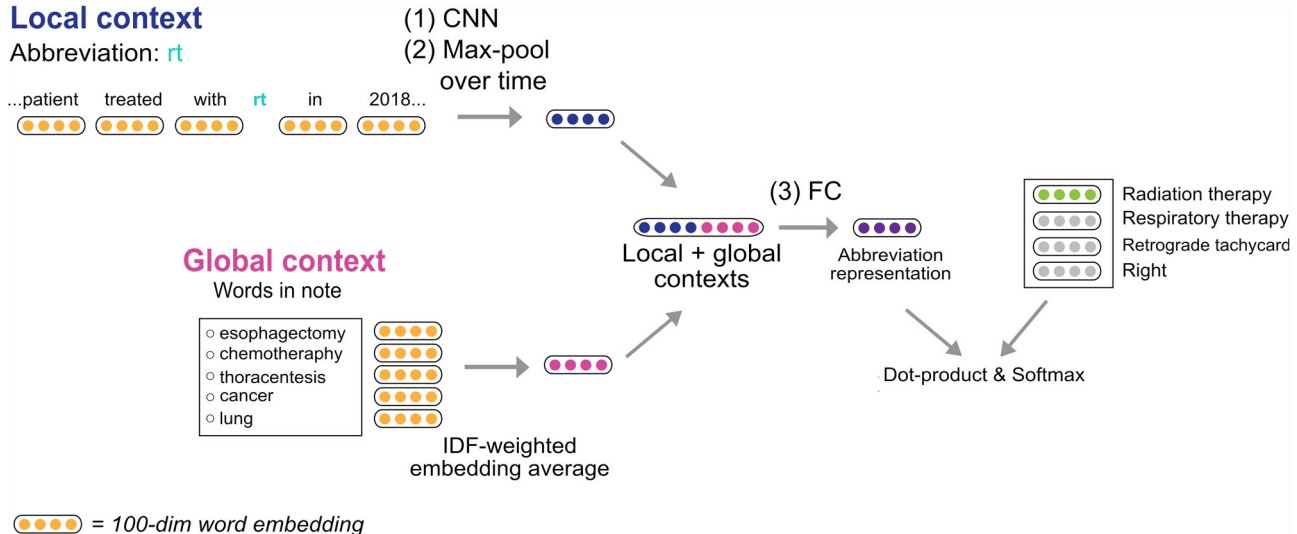

**Fig. 5 Overview of our abbreviation disambiguation model.** Sentences containing a target concept are passed through a convolutional neural network (CNN) and max-pooled over time to generate an encoding of the local context. Global context takes the IDF-weighted average of word embeddings in the entire document. We combine global context with the output from the sentence encoder and pass it through a fully connected layer (FC). We maximize the dot-product of the encoded sentence and expansion embedding.

*Pretraining.* To pre-train our model using structural relationships from UMLS, we first learned a 1-D CNN encoder similar to our abbreviation model that predicted what medical concept is present given its context. The embeddings for each concept were calculated by summing the raw embeddings of itself and its ancestors (as specified in the pre-training using ontological relationships section). We collected sentences for every concept in UMLS using the RS technique. To train our model, we used 1000 samples per concept and incorporated both a local context of 3 words and the global context. On average, the global context is 60 words per clinical note. We split the dataset into a training set (90%) and a validation set (10%). We used a learning rate of 0.002 and a batch size of 2048.

We initialized the weights of the convolution and fully connected layers with corresponding weights from the hierarchy model. If an expansion for a given abbreviation has a concept code in UMLS, we also initialized the expansion embedding in the abbreviation model with the corresponding embedding from the hierarchy.

**Reporting summary**. Further information on research design is available in the Nature Research Reporting Summary linked to this article.

## Data availability

The datasets used in this study are all publically available from the following sources: MIMIC III (https://physionet.org/content/mimiciii/1.4/, version 1.4). CASI (https://conservancy.umn.edu/handle/11299/137703). i2b2 (https://www.i2b2.org/NLP/DataSets/Main.php). UMLS (https://uts.nlm.nih.gov/uts/umls/home). AllAcronyms (https://www.allacronyms.com/_medical). We have published our code on how to generate training/validation/test sets from these datasets. Note that to make use of our code base, one must first obtain access to MIMIC III, i2b2, and/or UMLS. For the i2b2 hand-labeled dataset, we have noted the location of each abbreviation and its expansion in Supplementary Data 1. We are unable to provide clinical notes from The Hospital for Sick Children due to privacy restrictions.

## Code availability

Our code can be found at https://github.com/martaskrt/abbr_disamb.

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

## Acknowledgements

The authors would like to thank Nicole Sultanum for insightful discussions, as well as members of the SickKids Centre for Computational Medicine (CCM) and UHN DATA team for technical support, especially Pouria Mashouri and Rob Naccarato, and Mia Husic for her help preparing the manuscript.

## Author contributions

M.S. and M.B. conceptualized the methodology. M.S., M.B., A.A., and J.W. designed the experiments. M.S. preprocessed data carried out experiments and analyzed the results. E.D. compiled a dataset for the downstream clinical task and aided with statistical analysis. J.K. performed initial baseline experiments and created scripts for data collection. D.S. generated the i2b2 hand-labeled dataset and provided clinical expertise. M.S. and M.B. wrote the paper.

## Competing interests

The authors declare no competing interests.
