## [Peer Review File · Nature Communications]

Reviewers' Comments:

Reviewer #1:

Remarks to the Author:

This paper describes a novel approach toward the disambiguation of biomedical abbreviations using a combination of approaches, including methods integrating terminological resources to address the perennial problem of limited availability of annotated datasets, and evaluation of the utility of global contextual information as compared with the standard approach of considering local context only. As such, it addresses a fundamental problem in clinical Natural Language Processing (NLP), with strong improvements in performance demonstrated in multiple evaluation sets. The work is clearly presented, and the results support the authors' main claims in that the methods presented seem especially helpful in the case of abbreviations where training data are limited.

However, while some related literature is reviewed, the paper could be further strengthened by contextualizing the methods the authors have developed in relation to prior work in both the general and medical domain. In particular the method of integrating terminological knowledge seems closely related to the widely-used 'retrofitting' approach (Faraqui et al 2014), in which pre-trained word embeddings are retrofitted to a taxonomy by iteratively summing their original vectors with those representing their taxonomic neighbors. With respect to sentence embeddings, there is a rather broad literature on this topic in the general domain also. Could the authors please explain why they picked this particular method of generating sentence embeddings? Some additional discussion of the authors' methods in relation to this prior work would help the reader to identify the novel contributions of the current work.

The integration of global contextual information results in substantive improvements in performance, which is an important finding of the current work that also makes sense in relation to prevailing cognitive models of text comprehension. However, given the contemporary predominance of 'contextual' embedding methods (BERT and siblings) it is somewhat surprising to see that none of these already quite well established methods of encoding context were neglected in favor of a more straightforward superposition + concatenation approach. Did the authors consider using an attention mechanism (e.g. within a Transformer architecture) to encode context instead?

There are many different methods integrated to produce the performance boosts described in the work (e.g. RS+resampling, subword embeddings, vector postprocessing to embed taxonomic knowledge, data augmentation). As such, the paper would benefit from an overview and synthesis, perhaps in schematic form that describes each of these methods at a high level, and how the authors conceive of their contribution toward improved performance. A good starting point for this might be the last paragraph of the introduction, which enhancing RS for data augmentation (limited training data), leveraging structured relationships (rare terms?) and the addition of global context, but doesn't cover subword embeddings as another way of addressing the rare term problem.

Regarding the results, while the authors do point out the evaluations in which specific component methods lead to improvement, it isn't clear why for some of these methods this improvement would be inconsistent across datasets. For example, when taken alone global context improves performance over the Control method in two of four datasets only, and the same is true of SWR alone. While it is encouraging to see one of the combinations of these methods outperform the control in each of the sets, this doesn't give much insight into the strengths and the limitations of each of the component methods. Some additional analysis and discussion on why specific component methods are effective in some but not other reference sets would strengthen the paper. Furthermore, results with augmentation ('relatives') alone are not provided – could the authors please include them, or provide justification for their omission?

It seems as though there is a clear advantage for the methods presented on hand-labeled sets, and that this advantage is not apparent in the synthetically assembled RS sets. In this reviewer's opinion, the strength of the positive findings would be more readily apparent if these two sets of results were presented in separate sections and details, to the extent space requirements permit.

The difference in performance here may be more indicative of the limitations of RS-derived reference sets than the authors' methods, and while these limitations are discussed in this paper, clearly delineating the two evaluation paradigms would make it clearer to readers where they do or do not apply.

Presentation:

Results:

'with sentences containing closely related medical concepts determined using embedding distance' – this is ambiguous as both sentences and concepts are embedded. The Figure suggests it is the latter embeddings that are used, but this part of the paper is still quite hard to follow as it isn't obvious why augmenting an expansion with sentences from a closely related concept should work (these concepts may have different acronyms). It seems as though there is an unstated hypothesis that an acronym could serve as a replacement for an expansion of a related concept also, and spelling this out a bit would help the reader to follow the underlying rationale. It would also be helpful here to note in the text (rather than just in the diagram) that these augmented data serve as input for RS (at present the text suggests the sentences containing the related concepts without substitution are used to augment the training data).

'further details on each step is provided' – are provided

How should we interpret the first row of Table 1? Some of the entries are column headings, and others are results.

With Figure 2, it would be helpful to include the data set names in the Figure for ease of comparison with Table 1. Similarly, including 'hand-labelled' or 'RS' in each of the column headers of Table 1 would make the connection between these presentations of the results clearer. This does seem an important distinction to make, as the synthetic RS datasets obscure the utility of the authors' methods here.

Datasets – 'pre-train word embeddings using FastText and IDF weights' – could the authors please clarify how IDF weights were used to generate FastText embeddings? Or were these used to generate sentence embeddings downstream? Also, what hyperparameters were used when generating the word embeddings?

Regarding the use of the Euclidean distance – were the word/concept vectors normalized before this was calculated?

Pretraining – could the authors please provide additional details on the model 'similar to that developed by Arbab et al'? It would be preferable for the paper to stand alone, such that the details of what was done can be understood without the need to consult another paper.

Reviewer #2:

Remarks to the Author:

Developing a generalizable clinical abbreviation disambiguation system remains a challenge to clinical NLP. It is greatly appreciated that this study aims at addressing this challenging problem and shows promising results. The proposed approaches that consider related concepts and global features are useful. However, further investigation is needed to validate its true utility and assess the impact to current clinical NLP practice. Detailed comments include:

1. The contributions of three proposed new features (Relatives, Global, and HP) are not consistent across different datasets in Table 1: (1) the new methods achieved limited improvements on both datasets c and d; (2) it is good that large performance gains are obtained by new methods on datasets a and b; but they are not consistent, e.g., when global feature is added, performance increased on dataset a but dropped on dataset b; and (3) SWR, another baseline method, improved performance greatly on dataset b; but the same observation is not seen on dataset a. All these findings indicate that further investigation is necessary to better understand the behavior of

the proposed methods on different datasets, thus ensuring its generalizability.

2. It is my understanding that the entire system was built on MIMIC and UMLS and then applied to CASI, which serves as an independent test dataset. But in the "Experiments" section, authors stated that "We collected sentences containing expansions from CASI and medical concepts from UMLS using RS. In total, 105,161 concepts in UMLS were found in MIMIC-III." Why were CASI expanded concepts used in model training?

3. A question related to comment #2. One known issue about clinical abbreviation disambiguation is the completeness of sense inventories. Are all the abbreviations and their possible senses in CASI covered by those generated from MIMIC-III and UMLS? If not, please specify the coverage of the proposed RS approach on both abbreviations and their senses in the test datasets.

4. I would also suggest that do not remove any samples from the CASI dataset, so that authors can compare the performance between the proposed method and other previously published methods that used the same CASI dataset. In addition to comparing the proposed three new features with the "Control" baseline that is also developed by the authors, strong baselines of other published methods (e.g., other unsupervised methods) should be included. Authors should demonstrate that their methods are superior to the state-of-the-art unsupervised disambiguation methods.

5. The use of AllAcronyms need justification. The UMLS has abbreviations too. Does AllAcronyms have better coverage than UMLS?

6. Table 1, second row, headers for (b) and (d) need to be changed to "Macro" and "Micro".

Reviewer #3:

Remarks to the Author:

This manuscript describes an improved weakly supervised deep learning method for disambiguating medical abbreviation from clinical notes. The study is motivated as most existing methods generalize poorly on the samples from different distributions due to the absence of manually labeled training data for any specific abbreviation. As such, the authors propose a data augmentation technique that utilizes information from related UMLS medical concepts to automatically generate training data. And UMLS hierarchy relationships are being used to pre-train their proposed model. Moreover, weighted global context embeddings are incorporated into the model to further boost the performance. Experimental results on four test sets show that the proposed method improves the model's generalization ability with varying success, in comparison with their own baseline.

Major comments:

1. While abbreviation ambiguity is a well-known and widely studied problem for the clinical NLP community, it needs to be better introduced for the broader readership given the venue chosen here, with a focus on its current state of the art and remaining challenges. The authors have included several past studies, but a more complete background is desired given the amount of related work in the literature (especially in the last few years).

2. Although the manuscript cites several related studies (such as the references [4] [15] in the manuscript), none of them was used as baseline methods for comparison purposes. Instead, the superiority of the proposed method is demonstrated by comparing with itself – this is more like a feature ablation study. Direct comparisons with previous best-performing methods are much needed, in order to demonstrate the advancements to the state of the art.

3. From results in Table 1, one can see that most of the performance enhancements over the baselines come from data augmentation technique and global context information, which limits the impact of pre-training using ontological relationships. The model with pretraining (HP) only achieves very modest improvements on two of four test sets, but significant drops are observed on the i2b2 hand-labeled test set. The generalizability of this method may also be demonstrated in the same task but with a slightly different text genre (e.g. biomedical literature).

4. In addition to validate the validity of the proposed method, additional experiments should be considered to showcase the utility of this work. In particular, how could the work lead to significant improvements in downstream applications (e.g. entity linking or information extraction) in

processing clinical notes.

5. The manually annotated dataset appears to be limited, with only two dozens that are ambiguous. The annotations were done by medical students. Is there an annotation agreement on this small dataset?

6. The authors used FastText to generate word embeddings in this work. The FastText only allows a single context-independent representation for each word, but a word can have completely different semantics in different contexts. Instead, some widely used deep context-dependent representation methods should be considered, such as ELMo[1] and BERT[2]. The authors may also consider the potential value of incorporating contextualized embeddings into the model. And some related works (such as [3][4]) have shown that the contextualized embeddings can improve the performance of the model for medical abbreviation disambiguation.

[1] Peters ME, Neumann M, Iyyer M, et al. (2018). Deep contextualized word representations. In Proceedings of NAACL-HLT (pp. 2227-2237).

[2] Devlin J, Chang MW, Lee K, Toutanova K. (2019). BERT: Pre-training of Deep Bidirectional Transformers for Language Understanding. In Proceedings of NAACL-HLT (pp. 4171-4186).

[3] Jin Q, Liu J, Lu X. (2019). Deep Contextualized Biomedical Abbreviation Expansion. In Proceedings of the 18th BioNLP Workshop and Shared Task (pp. 88-96).

[4] Li I, Yasunaga M, Nuzumlalı MY, et al. (2019). A Neural Topic-Attention Model for Medical Term Abbreviation Disambiguation. arXiv preprint arXiv:1910.14076.

7. In evaluation, the proposed model was evaluated on four test sets. The authors provide the number of abbreviations in test sets excluding the MIMIC-III test set, and the number of abbreviations in the MIMIC-III test set should be provided. More importantly, why the results on three of four test sets are discussed (excluding the MIMIC-III test set)?

8. In the Experiment Section, sentences containing expansions from CASI and medical concepts from UMLS are collected to build the training data using reverse substitution. Can the sentences cover all abbreviations in four test sets? If an abbreviation cannot find any sentences containing its expansions from MIMIC-III, how is the training data created for the abbreviation?

9. A crowd-sourced database of abbreviations is used here. How significant is its role in this method? It is unclear in the current results.

10. Limitations of the proposed method should be properly discussed. For example, what is the remaining challenge in this case. Also, it appears that the current method needs to build one model for each abbreviation. There are ~30,000 medical abbreviations used in this work. How many models need to be developed in the training process? And what is the computational cost for model training and testing?

Minor comments:

in the formal discourse of textbooks and journal articles, an expansion is often provided for an abbreviation in its first occurrence. The authors noted that this is a much less common practice in clinical notes. Could the authors provide any statistics to illustrate this problem?

Page 4, the reference of i2b2 should be cited in the first occurrence.

In table 1, the table head (b) and (d) should be "Macro" and "Micro".

Many references have no title, such as [8] [9][10].

REVIEWER COMMENTS

Reviewer #1 (Remarks to the Author):

This paper describes a novel approach toward the disambiguation of biomedical abbreviations using a combination of approaches, including methods integrating terminological resources to address the perennial problem of limited availability of annotated datasets, and evaluation of the utility of global contextual information as compared with the standard approach of considering local context only. As such, it addresses a fundamental problem in clinical Natural Language Processing (NLP), with strong improvements in performance demonstrated in multiple evaluation sets. The work is clearly presented, and the results support the authors' main claims in that the methods presented seem especially helpful in the case of abbreviations where training data are limited.

Thank you for your feedback!

However, while some related literature is reviewed, the paper could be further strengthened by contextualizing the methods the authors have developed in relation to prior work in both the general and medical domain. In particular the method of integrating terminological knowledge seems closely related to the widely-used 'retrofitting' approach (Faraqui et al 2014), in which pre-trained word embeddings are retrofitted to a taxonomy by iteratively summing their original vectors with those representing their taxonomic neighbors. With respect to sentence embeddings, there is a rather broad literature on this topic in the general domain also. Could the authors please explain why they picked this particular method of generating sentence embeddings? Some additional discussion of the authors' methods in relation to this prior work would help the reader to identify the novel contributions of the current work.

Thank you for pointing out the work of Faraqui et al to us, we have referenced it in our paper on lines 457-460 as it is closely related to our hierarchical pretraining task. We chose our hierarchical pre-training task based on a study done by Arbabi et al (2019) because it showed improvements on clustering directly in the medical domain using the same ontology we use in this study. We picked this method of generating sentence embeddings for multiple reasons: (1) by using pretrained embeddings trained using FastText, we are able to predict embedding approximations for concepts that are not fully written out in the corpus (such as in vitro fertilization, in our case); (2) we can easily feed these embeddings into a CNN, which has been shown to produce state-of-the-art results for abbreviation disambiguation by Joopudi et al; (3) we can easily incorporate the global concept of the entire note using IDF-weighted embeddings.

The integration of global contextual information results in substantive improvements in performance, which is an important finding of the current work that also makes sense in relation to prevailing cognitive models of text comprehension. However, given the contemporary predominance of 'contextual' embedding methods (BERT and siblings) it is somewhat surprising to see that none of these already quite well established methods of encoding context were neglected in favor of a more straightforward superposition +

concatenation approach. Did the authors consider using an attention mechanism (e.g. within a Transformer architecture) to encode context instead?

That is an excellent point. The motivation behind our global context algorithm was that it is very straightforward. Computationally, it is neither time nor memory intensive and it shows a significant improvement. However, as you pointed out, it is important to consider state-of-the-art embedding techniques which may offer better contextualization within the global context. We repeated our experiments using ClinicalBERT and found that the performance was either the same or slightly worse than our baseline. The results can be viewed in the last row of Table 1 in the paper (right before line 201) and are described on lines 266-284 and 480-490. We think this may be because BERT learns very well from structured text, but in our case, these clinical notes are highly unstructured and so contextual embeddings provide no immediate benefit. However, we also find that when we incorporate our data augmentation technique, we still see a gain when using BERT (lines 280-284). This demonstrates that our upsampling technique is beneficial irrespective of the underlying model architecture.

There are many different methods integrated to produce the performance boosts described in the work (e.g. RS+resampling, subword embeddings, vector postprocessing to embed taxonomic knowledge, data augmentation). As such, the paper would benefit from an overview and synthesis, perhaps in schematic form that describes each of these methods at a high level, and how the authors conceive of their contribution toward improved performance. A good starting point for this might be the last paragraph of the introduction, which enhancing RS for data augmentation (limited training data), leveraging structured relationships (rare terms?) and the addition of global context, but doesn't cover subword embeddings as another way of addressing the rare term problem.

We agree with your comment and worked on rewriting the last paragraph of the introduction to explain each component of the pipeline in slightly more depth. The modifications can be viewed on lines 94-104.

Regarding the results, while the authors do point out the evaluations in which specific component methods lead to improvement, it isn't clear why for some of these methods this improvement would be inconsistent across datasets. For example, when taken alone global context improves performance over the Control method in two of four datasets only, and the same is true of SWR alone. While it is encouraging to see one of the combinations of these methods outperform the control in each of the sets, this doesn't give much insight into the strengths and the limitations of each of the component methods. Some additional analysis and discussion on why specific component methods are effective in some but not other reference sets would strengthen the paper. Furthermore, results with augmentation ('relatives') alone are not provided – could the authors please include them, or provide justification for their omission?

Thank you for pointing this out. The inconsistencies in the results were due to a copy-and-paste error (a header was missing, everything in column (b) and (d) in Table 1 should have been shifted down). The accurate results are clearly presented in Figure 2, which shows the same results as the (correct) Table 1 but using colours instead of numbers.

This error was fixed and the correct table is presented in the manuscript. With the correct table, the trend can be clearly seen that every component of our pipeline provides a benefit in all datasets.

We have included results with augmentation ('relatives') alone--these results can be viewed in Table 1.

It seems as though there is a clear advantage for the methods presented on hand-labeled sets, and that this advantage is not apparent in the synthetically assembled RS sets. In this reviewer's opinion, the strength of the positive findings would be more readily apparent if these two sets of results were presented in separate sections and details, to the extent space requirements permit. The difference in performance here may be more indicative of the limitations of RS-derived reference sets than the authors' methods, and while these limitations are discussed in this paper, clearly delineating the two evaluation paradigms would make it clearer to readers where they do or do not apply.

We agree that the advantages of our methods are not as visible in the RS-generated test sets as there are likely distributional shifts between concepts that are abbreviated and written out in their long form. To make the results table clearer, we created a heading that indicates the type of test set (hand-generated vs RS-generated) and wrote a few sentences at the beginning of the results section explaining these differences (lines 175-181).

Presentation:

Results:

'with sentences containing closely related medical concepts determined using embedding distance' – this is ambiguous as both sentences and concepts are embedded. The Figure suggests it is the latter embeddings that are used, but this part of the paper is still quite hard to follow as it isn't obvious why augmenting an expansion with sentences from a closely related concept should work (these concepts may have different acronyms). It seems as though there is an unstated hypothesis that an acronym could serve as a replacement for an expansion of a related concept also, and spelling this out a bit would help the reader to follow the underlying rationale. It would also be helpful here to note in the text (rather than just in the diagram) that these augmented data serve as input for RS (at present the text suggests the sentences containing the related concepts without substitution are used to augment the training data).

The key assumption is that sentences containing **long forms** of related terms (not abbreviation) can be used as proxies for unknown abbreviations. For example, sentences/notes which contain the phrase "assisted fertilization" are more similar to sentences that contain the term "ivf" meaning in vitro fertilization than to "ivf" meaning intravenous feeding, and hence can be used as training samples. We have now made it clearer in the text, lines 121-126.

'further details on each step is provided' – are provided

Thank you for pointing this out, this has been fixed.

How should we interpret the first row of Table 1? Some of the entries are column headings, and others are results.

Thank you for pointing this out. This was due to a copy-and-paste error; the rows should have been shifted down by one. This is now fixed and the table should read clearly.

With Figure 2, it would be helpful to include the data set names in the Figure for ease of comparison with Table 1. Similarly, including 'hand-labelled' or 'RS' in each of the column headers of Table 1 would make the connection between these presentations of the results clearer. This does seem an important distinction to make, as the synthetic RS datasets obscure the utility of the authors' methods here.

We agree with your suggestions. Both Table 1 and Figure 2 have been modified accordingly.

Datasets – 'pre-train word embeddings using FastText and IDF weights' – could the authors please clarify how IDF weights were used to generate FastText embeddings? Or were these used to generate sentence embeddings downstream? Also, what hyperparameters were used when generating the word embeddings?

The IDF weights were used to generate sentence embeddings for the global context. We generated the global context by taking an IDF-weighted average of the FastText embeddings. The motivation was that niche medical words would have a higher IDF weight and contribute more to the global context, while common words would contribute less.

To train the FastText model, we used a skip-gram architecture and kept all default hyperparameters used in the FastText documentation. We have added additional clarification to the text describing this (line 515-517).

Regarding the use of the Euclidean distance – were the word/concept vectors normalized before this was calculated?

No, we did not normalize the word vectors prior to calculating the Euclidean distance between medical concepts. It has been argued that the length of a word vector encodes term frequency as well as information about context similarity (Schakel and Wilson 2015, <https://arxiv.org/pdf/1508.02297.pdf>). For example, words represented by longer vectors refer to unique contexts. We thought this would be better for determining the relatedness of medical concepts, although we did not test this experimentally.

Pretraining – could the authors please provide additional details on the model 'similar to that developed by Arbabi et al'? It would be preferable for the paper to stand alone, such that the details of what was done can be understood without the need to consult another paper.

Thank you for pointing this out. We replaced this with a few sentences that explain more precisely what the training procedure is on lines 526-529.

Reviewer #2 (Remarks to the Author):

Developing a generalizable clinical abbreviation disambiguation system remains a challenge to clinical NLP. It is greatly appreciated that this study aims at addressing this challenging problem and shows promising results. The proposed approaches that consider related concepts and global features are useful. However, further investigation is needed to validate its true utility and assess the impact to current clinical NLP practice. Detailed comments include:

1. The contributions of three proposed new features (Relatives, Global, and HP) are not consistent across different datasets in Table 1: (1) the new methods achieved limited improvements on both datasets c and d; (2) it is good that large performance gains are obtained by new methods on datasets a and b; but they are not consistent, e.g., when global feature is added, performance increased on dataset a but dropped on dataset b; and (3) SWR, another baseline method, improved performance greatly on dataset b; but the same observation is not seen on dataset a. All these findings indicate that further investigation is necessary to better understand the behavior of the proposed methods on different datasets, thus ensuring its generalizability.

Thank you for pointing this out. The inconsistencies in the results were due to a copy-and-paste error (a header was missing, everything in column (b) and (d) in Table 1 should have been shifted down). The accurate results are clearly presented in Figure 2, which shows the same results as the (correct) Table 1 but using colours instead of numbers. This error was fixed and the correct table is presented in the manuscript. With the correct table, the trend can be clearly seen that every component of our pipeline provides a benefit in all datasets.

2. It is my understanding that the entire system was built on MIMIC and UMLS and then applied to CASI, which serves as an independent test dataset. But in the “Experiments” section, authors stated that “We collected sentences containing expansions from CASI and medical concepts from UMLS using RS. In total, 105,161 concepts in UMLS were found in MIMIC-III.” Why were CASI expanded concepts used in model training?

Thank you for pointing this out, our explanation needed to be clearer. You are correct in that the entire system was trained on MIMIC and UMLS. For fair comparison between the baseline model and the model with our data augmentation technique, we constrict the possible expansions our model can predict to only those that are present in the target distribution. This is because using our data augmentation technique, we can generate training examples for any class (i.e. everything in AllAcronyms), whereas the baseline model can only generate training samples for concepts that are written in the text (i.e. concepts only in MIMIC). We thus constrained the set of possible expansions to those available to all methods, in order to allow for direct comparisons.

3. A question related to comment #2. One known issue about clinical abbreviation disambiguation is the completeness of sense inventories. Are all the abbreviations and their possible senses in CASI covered by those generated from MIMIC-III and UMLS? If not, please specify the coverage of the proposed RS approach on both abbreviations and their senses in the test datasets.

You are correct in that the overlap of the sense inventories is not perfect. Because we generated training/test samples from MIMIC and i2b2 using RS using our list of abbreviations and expansions from AllAcronyms, there are no out-of-distribution expansions in the test set (although we are not able to generate test samples for expansions in AllAcronyms that are also not present in MIMIC-III, so the number of expansions we test on is smaller than the number of expansions in AllAcronyms). For the hand-labelled datasets (CASI and i2b2), this is not the case because the true expansions in those datasets may not be present in AllAcronyms. The number of expansions in CASI not present in the expansion list from AllAcronyms is 107/250 total (42.8%). The number of abbreviations in CASI that are not present in AllAcronyms is 5/65 (7.7%). The number of handlabelled expansions in i2b2 not present in the expansion list from AllAcronyms is 10/53 total (18.9%). Since we generated a list of potential abbreviations for handlabelling based on the abbreviation list in AllAcronyms, there is perfect coverage of abbreviations.

4. I would also suggest that do not remove any samples from the CASI dataset, so that authors can compare the performance between the proposed method and other previously published methods that used the same CASI dataset. In addition to comparing the proposed three new features with the “Control” baseline that is also developed by the authors, strong baselines of other published methods (e.g., other unsupervised methods) should be included. Authors should demonstrate that their methods are superior to the state-of-the-art unsupervised disambiguation methods.

Unfortunately there is no way to make the results for the CASI dataset directly comparable as previous work that tested on CASI removed around 200 samples due to spelling mistakes and other issues but didn't indicate which samples (Joopudi et al 2018). Thus, although we know which abbreviations were evaluated in previous work, we don't know which samples exactly were kept in the dataset.

It has been previously found in Joopudi et al 2018 that CNNs produce better results for abbreviation disambiguation compared to other models such as SVMs or KNNs, which is why we opted for CNN architecture. Although we don't write about other methods in this paper, this has been analyzed in previous work (Joopudi et al 2018, I Li 2019). Furthermore, we included results from BERT in this paper which show that incorporating our methods (specifically, the data augmentation technique for poorly-represented medical concepts) improve results agnostic to the baseline model architecture.

5. The use of AllAcronyms need justification. The UMLS has abbreviations too. Does AllAcronyms have better coverage than UMLS?

AllAcronyms is a citizen science platform where users can upload abbreviations and their expansions across multiple domains, including medicine. After cleaning, AllAcronyms has 30,974 abbreviations (cleaning process described in Methods on lines 349-350). Out of these abbreviations, 12,827 exist in UMLS. There may be abbreviations in UMLS that do not appear in AllAcronyms, but this is nontrivial to determine concepts are not labelled as abbreviations in UMLS as far as we know.

6. Table 1, second row, headers for (b) and (d) need to be changed to “Macro” and “Micro”.

Thank you for pointing this out. This was due to a copy-and-paste error; the rows should have been shifted down by one. This is now fixed and the table should read clearly.

Reviewer #3 (Remarks to the Author):

This manuscript describes an improved weakly supervised deep learning method for disambiguating medical abbreviation from clinical notes. The study is motivated as most existing methods generalize poorly on the samples from different distributions due to the absence of manually labeled training data for any specific abbreviation. As such, the authors propose a data augmentation technique that utilizes information from related UMLS medical concepts to automatically generate training data. And UMLS hierarchy relationships are being used to pre-train their proposed model. Moreover, weighted global context embeddings are incorporated into the model to further boost the performance. Experimental results on four test sets show that the proposed method improves the model's generalization ability with varying success, in comparison with their own baseline.

Major comments:

1. While abbreviation ambiguity is a well-known and widely studied problem for the clinical NLP community, it needs to be better introduced for the broader readership given the venue chosen here, with a focus on its current state of the art and remaining challenges. The authors have included several past studies, but a more complete background is desired given the amount of related work in the literature (especially in the last few years).

Thank you for the suggestion. We have expanded the introduction, adding new citations to several previous papers to provide more historical background to the problem, including Liu et al (2002) and Pakhamov et al (2005) (lines 44-51), as well as more recent advancements in language modelling (lines 56-58). We also included a citation to Faraqi et al as suggested by reviewer 1 and have added more text to the discussion explaining the limitations of our work and the remaining challenges (lines 316-327).

2. Although the manuscript cites several related studies (such as the references [4] [15] in the manuscript), none of them was used as baseline methods for comparison purposes. Instead, the superiority of the proposed method is demonstrated by comparing with itself – this is more like a feature ablation study. Direct comparisons with previous best-performing methods are much needed, in order to demonstrate the advancements to the state of the art.

It has been previously found in several studies (Joopudi et al 2018, Li 2019) that CNNs produce better results for abbreviation disambiguation compared to other models such as SVMs or KNNs, which is why we opted for CNN architecture. Furthermore, we included results from BERT in this paper which show that incorporating our methods (specifically, the data augmentation technique for poorly-represented medical concepts) improve results agnostic to the baseline model architecture.

While a direct comparison to previous published methods would indeed be preferable, these methods typically use cross-validation on the same dataset to train the method, rather than an external one, use private datasets or remove specific entries from public datasets,

making such direct comparisons impossible (Joopudi et al 2018, Finley et al 2016). As far as we know, the code used in the papers is not available.

3. From results in Table 1, one can see that most of the performance enhancements over the baselines come from data augmentation technique and global context information, which limits the impact of pre-training using ontological relationships. The model with pretraining (HP) only achieves very modest improvements on two of four test sets, but significant drops are observed on the i2b2 hand-labeled test set. The generalizability of this method may also be demonstrated in the same task but with a slightly different text genre (e.g. biomedical literature).

Thank you for pointing this out. The inconsistencies in the results were due to a copy-and-paste error (a header was missing, everything in column (b) and (d) in Table 1 should have been shifted down). The accurate results are clearly presented in Figure 2, which shows the same results as the (correct) Table 1 but using colours instead of numbers. This error was fixed and the correct table is presented in the manuscript. With the correct table, the trend can be clearly seen that every component of our pipeline provides a benefit in all datasets.

4. In addition to validate the validity of the proposed method, additional experiments should be considered to showcase the utility of this work. In particular, how could the work lead to significant improvements in downstream applications (e.g. entity linking or information extraction) in processing clinical notes.

Thank you for the suggestion. While our method significantly outperforms the baselines, we have struggled to identify a downstream application where this impact will be directly visible. Entity linking and information extraction from a clinical note utilizes a lot of information in the note besides acronyms, so any residual difference due to a change in the disambiguation method would be small.

We acquired clinical notes from the Hospital for Sick Children that contain a list of symptoms the patients experienced as well as the tests they ended up receiving. Improving workflows in emergency departments by triaging patients to receiving appropriate tests is currently an area being explored in Machine Learning for Healthcare. We found that disambiguating abbreviations in clinical notes before training a prediction model improves the final performance from 78.09% to 78.51% ($p < 0.05$). We briefly describe this result in our paper (lines 286-297, 492-508), however a full analysis will be published elsewhere.

5. The manually annotated dataset appears to be limited, with only two dozens that are ambiguous. The annotations were done by medical students. Is there an annotation agreement on this small dataset?

Unfortunately the manually-annotated dataset is indeed limited, but this is not entirely surprising, as i2b2 is a smaller dataset (40 Mb after cleaning vs 1.7 GB in MIMIC after cleaning) and from a niche discipline. Most of the sentences were labelled by 2 students; any discrepancies were resolved by our partner clinician. Although we had a number of

candidate abbreviations to present to the hand labellers, we found that the overwhelming majority (90%) only had one expansion, meaning we could not evaluate our model on it.

6. The authors used FastText to generate word embeddings in this work. The FastText only allows a single context-independent representation for each word, but a word can have completely different semantics in different contexts. Instead, some widely used deep context-dependent representation methods should be considered, such as ELMo[1] and BERT[2]. The authors may also consider the potential value of incorporating contextualized embeddings into the model. And some related works (such as [3][4]) have shown that the contextualized embeddings can improve the performance of the model for medical abbreviation disambiguation.

[1] Peters ME, Neumann M, Iyyer M, et al. (2018). Deep contextualized word representations. In Proceedings of NAACL-HLT (pp. 2227-2237).

[2] Devlin J, Chang MW, Lee K, Toutanova K. (2019). BERT: Pre-training of Deep Bidirectional Transformers for Language Understanding. In Proceedings of NAACL-HLT (pp. 4171-4186).

[3] Jin Q, Liu J, Lu X. (2019). Deep Contextualized Biomedical Abbreviation Expansion. In Proceedings of the 18th BioNLP Workshop and Shared Task (pp. 88-96).

[4] Li I, Yasunaga M, Nuzumlalı MY, et al. (2019). A Neural Topic-Attention Model for Medical Term Abbreviation Disambiguation. arXiv preprint arXiv:1910.14076.

We repeated our experiments using ClinicalBERT and found that the performance was either the same or slightly worse than our baseline. The results can be viewed in the last row of Table 1 in the paper (right before line 201) and are described on lines 266-284 and 480-490. We think this may be because BERT learns very well from structured text, but in our case, these clinical notes are highly unstructured and so contextual embeddings provide no immediate benefit, while the added complexity of the model makes it difficult to train. However, we also find that when we incorporate our data augmentation technique, we still see a gain when using BERT (lines 280-284). This demonstrates that our upsampling technique is beneficial irrelevant of the underlying model architecture.

7. In evaluation, the proposed model was evaluated on four test sets. The authors provide the number of abbreviations in test sets excluding the MIMIC-III test set, and the number of abbreviations in the MIMIC-III test set should be provided. More importantly, why the results on three of four test sets are discussed (excluding the MIMIC-III test set)?

Thank you for pointing this out, the number of abbreviations in the MIMIC-III test set was 1116, this has now been added into the main text on lines 362-364. We mainly use this test set as a sanity check to ensure our model performs well within the distribution it was trained on and to demonstrate that the typical training procedure overfits to the MIMIC distribution. As expected, on this test set our data augmentation techniques do not improve performance much: RS is a direct match to the test dataset, and data augmentation injects noise into the model, which reduces overfitting but does not improve performance.

8. In the Experiment Section, sentences containing expansions from CASI and medical concepts from UMLS are collected to build the training data using reverse substitution. Can

the sentences cover all abbreviations in four test sets? If an abbreviation cannot find any sentences containing its expansions from MIMIC-III, how is the training data created for the abbreviation?

As expanded above in response to reviewer 2, the set of senses does not have perfect overlap. However if an expansion cannot be found in MIMIC-III, we are still able to predict an embedding for it using the FastText model as it takes apart the word and combines embeddings using n-gram information. Using this predicted embedding, we can find nearby related medical concepts and generated the training set using those (it's a noisy approximation).

9. A crowd-sourced database of abbreviations is used here. How significant is its role in this method? It is unclear in the current results.

The crowd-sourced database was used for two reasons: (1) as motivation to highlight potentially how many abbreviations exist, (2) to generate a list of abbreviations and expansions that was then cleaned by matching it with concepts in UMLS. This was significant for creating the MIMIC training and test sets, and i2b2 test set.

Note that in the final model evaluation, for fair comparison between the baseline model and the model with our data augmentation technique, we constrict the possible expansions our model can predict to only those that are present in the target distribution. This is because using our data augmentation technique, we can generate training examples for any class (i.e. everything in AllAcronyms), whereas the baseline model can only generate training samples for concepts that are written in the text (i.e. concepts only in MIMIC). This increases the number of possible classes that can be predicted significantly, which is not fair for head-to-head comparison of the models.

10. Limitations of the proposed method should be properly discussed. For example, what is the remaining challenge in this case. Also, it appears that the current method needs to build one model for each abbreviation. There are ~30,000 medical abbreviations used in this work. How many models need to be developed in the training process? And what is the computational cost for model training and testing?

Thank you for the suggestion and questions. We have added the following to the discussion to address these points.

While our work showed an improvement from 67 to 84% relative to the baseline on the CASI dataset when trained on MIMIC, the performance does not match the performance of classifiers directly trained on CASI via cross-validation (>95%, Joopudi et al 2018). Thus a significant gap still remains for methodological improvement to allow for fully generalizable abbreviation disambiguation.

As was done in previous work (Joopudi et al 2018, Finley et al 2016), we train one model per abbreviation, and the training cost is significantly more expensive using our data augmentation technique than the baseline model since we do 25 rounds of Bayesian optimization to search for the optimal temperature (although this can likely be reduced, but we did not experiment with that in this study). To give a concrete example, training 65

abbreviations (the size of CASI dataset) on a single Tesla V-100 GPU takes ~25 hours using our data augmentation technique but ~1 hour for the baseline.

We have now added these points to the discussion (lines 316-327).

Minor comments:

in the formal discourse of textbooks and journal articles, an expansion is often provided for an abbreviation in its first occurrence. The authors noted that this is a much less common practice in in clinical notes. Could the authors provide any statistics to illustrate this problem?

Unfortunately we do not have access to any such statistics; in our experience with clinical notes in multiple hospitals we have never seen abbreviations defined at first use due to lack of time. Often they are only used once in the whole note.

Page 4, the reference of i2b2 should be cited in the first occurrence.

In table 1, the table head (b) and (d) should be “Macro” and “Micro”.

Many references have no title, such as [8] [9][10].

Thank you for pointing these out! These are all fixed.

Reviewers' Comments:

Reviewer #1:

Remarks to the Author:

This is the first revision of a manuscript describing some innovative approaches to augmenting datasets for the disambiguation of biomedical acronyms. The authors have addressed many of the concerns raised upon review of the original edition, and have included results using ClinicalBERT as a point of comparison, cleaning up of a critical error in one of the tables, as well as some additional experiments to evaluate utility in a downstream task.

The question arises as to whether or not inclusion of ClinicalBERT alone addresses the concern raised by multiple reviewers that no baseline models were provided as points of comparison. While I'm not sure I agree with all of the authors' justifications for not including certain models (the code from Finley et al appears to be publicly available here https://github.com/gpfinley/towards_comprehensive), I appreciate the new finding that their data augmentation technique produces gains across two models (albeit in only one dataset so far). In fact I would argue this finding is sufficiently important that it should feature in the results table (Table 1), along with the unaugmented ClinicalBERT model. Doing so would require applying this technique to BERT with the i2b2 and Mimic data sets also, or justifying some empty cells if this is impractical on account of computational constraints. While it would be interesting to see if this assertion holds with another baseline model, I'm not sure more baselines are strictly necessary to support the main claims of the paper because these have to do with the utility of data augmentation rather than whether or not the underlying model is state of the art.

Could the authors please clarify the relationship between the current work and their paper Health NeurIPS Workshop, PMLR 116:233-245, 2020? I understand that this is a workshop proceedings paper, not a journal paper – but it seems as though the current paper should acknowledge this work and explain how it extends it.

There is a sentence fragment ("Recently,") at the end of the second paragraph on page 27.

Reviewer #4:

Remarks to the Author:

As advised, I have focused my review on the changes in response to the comments of the original Reviewer #3.

The authors have responded to the concerns related to lack of comparison with BERT-based methods and the impact of the abbreviation expansion on a downstream task by adding additional experiments. While the downstream task experiment in particular deserves more space to unpack the results, I understand and accept the authors' choice to leave it as indicative in this context.

The comment about inter-annotator agreement on the i2b2 manually-annotated dataset has not been directly addressed. While the response indicates that the data was doubly annotated and that an adjudication step was applied, it remains unclear what the agreement between the two annotators was in the first step. Only 24 abbreviations had multiple expansions, but it is unclear how many sentences this corresponds to or how many cases required adjudication. This is relevant information about the data set that should be added. This is, however, complicated by the use of 7 individual annotators.

Other comments have been adequately addressed.

Please check and correct where relevant any arXiv references to the peer-reviewed publication. For instance Jin et al (2019) should refer to:

<https://www.aclweb.org/anthology/W19-5010/> and Faroqui et al (2015) should refer to:

<https://www.aclweb.org/anthology/N15-1184/> [There are others that similarly need to be corrected.] arXiv references should only be used where no formal publication has appeared. Don't

just use the default citation details from Google Scholar, which tends to be the first appearance of a paper rather than the official publication.

REVIEWERS' COMMENTS

Reviewer #1 (Remarks to the Author):

This is the first revision of a manuscript describing some innovative approaches to augmenting datasets for the disambiguation of biomedical acronyms. The authors have addressed many of the concerns raised upon review of the original edition, and have included results using ClinicalBERT as a point of comparison, cleaning up of a critical error in one of the tables, as well as some additional experiments to evaluate utility in a downstream task.

The question arises as to whether or not inclusion of ClinicalBERT alone addresses the concern raised by multiple reviewers that no baseline models were provided as points of comparison. While I'm not sure I agree with all of the authors' justifications for not including certain models (the code from Finley et al appears to be publicly available here https://github.com/gpfinley/towards_comprehensive),

Thank you for linking this repository. We have run our dataset through this code and find that our baseline model ("Control") has a 3% improvement (64% Finley vs 67.2% us) on the CASI dataset. This is now added to the manuscript.

I appreciate the new finding that their data augmentation technique produces gains across two models (albeit in only one dataset so far). In fact I would argue this finding is sufficiently important that it should feature in the results table (Table 1), along with the unaugmented ClinicalBERT model. Doing so would require applying this technique to BERT with the i2b2 and Mimic data sets also, or justifying some empty cells if this is impractical on account of computational constraints. While it would be interesting to see if this assertion holds with another baseline model, I'm not sure more baselines are strictly necessary to support the main claims of the paper because these have to do with the utility of data augmentation rather than whether or not the underlying model is state of the art.

We have additionally run our data augmentation techniques on the i2b2 hand-labelled dataset on the ClinicalBERT model and found a 9% improvement compared to the baseline ClinicalBERT model. We have included these findings in Table 1. It is computationally impractical to run these techniques with ClinicaBERT on the reverse substitution-generated datasets so we have omitted them from the table.

Could the authors please clarify the relationship between the current work and their paper Health NeurIPS Workshop, PMLR 116:233-245, 2020? I understand that this is a workshop proceedings paper, not a journal paper – but it seems as though the current paper should acknowledge this work and explain how it extends it.

Absolutely. We have noted our previous work in the discussion section, and also added an explanation of the key extensions introduced in this paper:

Our approach is based on the ideas introduced by us in an extended conference abstract (Skreta et al. 2020), where we explored global context and data augmentation to improve

performance for this task. This manuscript, however, goes beyond the preliminary work by introducing the use of hierarchical ontologies as a pretraining step, improving the Bayesian Optimization algorithm to select optimal temperature, creating and analyzing the hand-labelled dataset as another orthogonal metric, tripling the set of analyzed abbreviations, and enhancing the Results section by including the comparison to Clinical BERT, and Finley et al, as well as including the analysis of the downstream clinical task (test prediction).

There is a sentence fragment (“Recently,”) at the end of the second paragraph on page 27.

Thank you for pointing this out; it has been removed.

Reviewer #4 (Remarks to the Author):

As advised, i have focused my review on the changes in response to the comments of the original Reviewer #3.

The authors have responded to the concerns related to lack of comparison with BERT-based methods and the impact of the abbreviation expansion on a downstream task by adding additional experiments. While the downstream task experiment in particular deserves more space to unpack the results, I understand and accept the authors' choice to leave it as indicative in this context.

The comment about inter-annotator agreement on the i2b2 manually-annotated dataset has not been directly addressed. While the response indicates that the data was doubly annotated and that an adjudication step was applied, it remains unclear what the agreement between the two annotators was in the first step. Only 24 abbreviations had multiple expansions, but it is unclear how many sentences this corresponds to or how many cases required adjudication. This is relevant information about the data set that should be added. This is, however, complicated by the use of 7 individual annotators.

Thank you for pointing this out. We have a table in the Supplementary Information (Table S1) that indicates the exact number of sentences for every expansion for every abbreviation. We have included a reference to this in the manuscript. The agreement between multiple annotators was fairly consistent, with only 37 disagreements out of the 4253 total sentences labelled. We had at least two annotators per sentence, although all 7 annotators did not see every sentence. Our clinical collaborator resolved any ambiguous cases (210/4253) and checked over the entire final dataset before we tested our model on it.

Other comments have been adequately addressed.

Please check and correct where relevant any arXiv references to the peer-reviewed publication. For instance Jin et al (2019) should refer to: <https://www.aclweb.org/anthology/W19-5010/> and Faroqui et al (2015) should refer to: <https://www.aclweb.org/anthology/N15-1184/> [There are others that similarly need to be corrected.] arXiv references should only be used where no formal publication has appeared.

Don't just use the default citation details from Google Scholar, which tends to be the first appearance of a paper rather than the official publication.

Thank you for bringing this to our attention. We believe that we have now correctly formatted all citations.